# Integrated Indoor Positioning System of Greenhouse Robot Based on UWB/IMU/ODOM/LIDAR

**DOI:** 10.3390/s22134819

**Published:** 2022-06-25

**Authors:** Zhenhuan Long, Yang Xiang, Xiangming Lei, Yajun Li, Zhengfang Hu, Xiufeng Dai

**Affiliations:** College of Mechanical and Electrical Engineering, Hunan Agriculture University, Changsha 410128, China; lzh20190092@163.com (Z.L.); lxm@stu.hunau.edu.cn (X.L.); lyj202120001@163.com (Y.L.); victoria@stu.hunau.edu.cn (Z.H.); dxf@stu.hunau.edu.cn (X.D.)

**Keywords:** greenhouse, robots, UWB/IMU/ODOM/LIDAR, indoor positioning

## Abstract

Conventional mobile robots employ LIDAR for indoor global positioning and navigation, thus having strict requirements for the ground environment. Under the complicated ground conditions in the greenhouse, the accumulative error of odometer (ODOM) that arises from wheel slip is easy to occur during the long-time operation of the robot, which decreases the accuracy of robot positioning and mapping. To solve the above problem, an integrated positioning system based on UWB (ultra-wideband)/IMU (inertial measurement unit)/ODOM/LIDAR is proposed. First, UWB/IMU/ODOM is integrated by the Extended Kalman Filter (EKF) algorithm to obtain the estimated positioning information. Second, LIDAR is integrated with the established two-dimensional (2D) map by the Adaptive Monte Carlo Localization (AMCL) algorithm to achieve the global positioning of the robot. As indicated by the experiments, the integrated positioning system based on UWB/IMU/ODOM/LIDAR effectively reduced the positioning accumulative error of the robot in the greenhouse environment. At the three moving speeds, including 0.3 m/s, 0.5 m/s, and 0.7 m/s, the maximum lateral error is lower than 0.1 m, and the maximum lateral root mean square error (RMSE) reaches 0.04 m. For global positioning, the RMSEs of the x-axis direction, the y-axis direction, and the overall positioning are estimated as 0.092, 0.069, and 0.079 m, respectively, and the average positioning time of the system is obtained as 72.1 ms. This was sufficient for robot operation in greenhouse situations that need precise positioning and navigation.

## 1. Introduction

Stable and reliable absolute positioning information can be obtained by global navigation satellite system (GNSS) in the outdoor environment [1], whereas under the indoor conditions of greenhouses, it is impossible to apply GNSS for accurate positioning due to the signal occlusion. At present, the positioning and navigation methods of greenhouse robots worldwide primarily consist of guide rail, machine vision navigation, ultrasonic navigation, and multi-sensor fusion navigation [2,3,4,5]. Feng Qingchun et al. [6] designed a greenhouse robot to walk autonomously and pick tomatoes on the set track. This positioning greenhouse robot has high accuracy and reliability and can achieve centimeter-level positioning. However, specific tracks should be set in advance, which is costly and inflexible [7]. Li Tianhua et al. [8] proposed controlling the direction of view of the camera through the pan-tilt so the visual axis of the camera could always be parallel to the road. Moreover, machine vision was adopted to extract the pixel coordinates of the horizontal center point at the end of the road to obtain navigation information, as an attempt to achieve navigation. The maximum deviation of using the telephoto camera in the straight-line driving of the greenhouse should be lower than 0.15 m. The navigation method based on machine vision is capable of achieving more accurate path tracking, whereas it is difficult to maintain the stability of the positioning system in the greenhouse since it is dependent on good lighting conditions. Mosalanejad et al. [9] designed a spray robot to measure the distance of obstacles with ultrasonic sensors, and this robot could autonomously walk in the greenhouse. The lateral Root Mean Square Error (RMSE) was found to be less than 0.08 m at different driving speeds. The method is only dependent on a single ultrasonic sensor to locate and navigate the robot, which is susceptible to environmental interference and has poor flexibility. LIDAR, a novel type of navigation and positioning sensor, has become the mainstream sensor of robot navigation for its advantages of high stability, high precision, and high instantaneity [10,11]. Hou Jialin et al. [12] used the integration of front and rear dual-lidar and wheel encoder based on Simultaneous Localization and Mapping (SLAM) algorithm to achieve the global positioning, mapping, and navigation functions of the robot in the greenhouse. The RMSE was found to be less than 0.11 m between the actual path and the target path of the on-board system in the greenhouse constant speed navigation, and the horizontal and vertical RMSE of the target point navigation was measured to be less than 0.12 m. However, this method easily results in positioning accumulative error after long-time work. In general, the global positioning method based on LIDAR should integrate multiple positioning information for the accurate positioning and navigation of the robot. At present, IMU/ODOM/LIDAR integrated method has been the most extensively used by indoor robots, whereas it is easy to produce accumulative error caused by wheel slip under complicated ground conditions in greenhouse [13,14]. In addition, the structured site of the greenhouse and the mass planting of the same crops lead to highly similar environments, which may cause the problem of LIDAR scanning matching failure. As a result, the probability of kidnapped robots problem in global positioning increases [15], and the positioning and navigation accuracy of the greenhouse robot is affected. Therefore, introducing a sensor capable of providing absolute location information can effectively decrease the probability of the above problem. Ultra-wideband (UWB) technology, because of its relatively high absolute positioning accuracy, has become a positioning system employed by indoor mobile robots to determine absolute coordinates instead of outdoor GNSS positioning [16]. The UWB positioning system platform is reasonably built in an open environment which is capable of achieving decimeter-level positioning, and it is characterized by strong real-time data, no accumulative error, and high positioning accuracy. It applies to the positioning and tracking of indoor static or dynamic objects, whereas the data of its ranging method based on signal arrival time fluctuates [17]. Lin Xiangze et al. [18] introduced UWB positioning into agricultural vehicles in greenhouses, solved ranging error by k-means algorithm and truncation processing method, and obtained accurate positioning information. The average static positioning accuracy is 0.07 m, the probability of dynamic accuracy reaching 0.08 m is 31.3%, and the probability of higher than 0.15 m is 35%. This method adopts UWB as the only location information source, so it is difficult to provide positioning information with high continuity and high stability.

To summarize, in the greenhouse environment, the conventional method of indoor global positioning and navigation for mobile robots using LIDAR suffers from the problems of low mapping and positioning accuracy caused by positioning accumulative error, whereas UWB positioning can provide global positioning information with high precision, no accumulative error, and strong real-time. A positioning system based on UWB/IMU/ODOM/LIDAR, which can correct the greenhouse robot's accumulated positioning error in global positioning, is developed in this paper. This system can present accurate and stable positioning information to the greenhouse robot, which is the goal of this research.

## 2. Materials and Methods

### 2.1. Composition and Design of Positioning System

The integrated positioning system in this paper primarily consists of an UWB positioning system, a robot platform, and a remote monitoring platform, as presented in Figure 1. To be specific, the UWB positioning system consists of four base stations (including base stations A, B, C, and D), as well as a positioning label. In the robot platform, four-wheel drive and differential steering are adopted. The respective wheel is equipped with a shock absorber, thus effectively reducing the odometer (ODOM) error arising from wheel slip under the complex ground conditions of the greenhouse. The robot platform is equipped with a UWB label, a LIDAR, an inertial measurement unit (IMU), as well as four photoelectric encoders. The remote monitoring platform employs a notebook computer to achieve the communication and remote monitoring of the robot platform via Wi-Fi.

The UWB positioning system adopts D-DWG-PG 2.5 positioning module designed by Guangzhou LENET Technology Co., Ltd. (Guangzhou, China). The maximum communication distance between modules is 130 m. In this paper, four modules serve as the base stations, and the square positioning area is built. The distance between UWB positioning label and the four base stations is measured by time of flight (TOF) method, and then the absolute positioning information of the robot platform is solved by triangulation algorithm [17]. This platform also adopts the RPLIDAR-S2 2D LIDAR of Silan technology to acquire the point cloud information of the environment. The LIDAR adopts TOF for ranging, with excellent light resistance and high measurement accuracy. It has 360° horizontal scanning range, 0.12° horizontal angle resolution, 0.05~30 m measurement radius, ±3 cm measurement accuracy, and 15 Hz scanning frequency. It is capable of efficiently acquiring the 2D environmental information of the greenhouse, thus becoming suitable for all-weather operation of robot platform. The LIDAR is installed on the top of the robot platform, which scans 360° environmental information. IMU employs iFLYTEK's WHEELTEC-N100 9-axis attitude sensor to acquire yaw information, with an angular resolution of 0.1°. To be specific, it is installed on the top of the robot platform and located at the rotation center of the robot platform. Four photoelectric encoders are installed on four motor drive wheels, respectively, to serve as an ODOM for providing speed and mileage information. The robot platform adopts STM32 module serves as the lower computer to control the DC motor to drive the wheels, and the driving speed of the robot platform is monitored and fed back via the photoelectric encoder. The robot platform adopts Raspberry Pi 4B installed with Ubuntu (18.04) and ROS (Melodic) system as the upper computer. On the basis of ROS(Melodic), this platform achieves the positioning and navigation algorithm, sensor data monitoring, and sending speed control instructions to the lower computer [19].

### 2.2. Integrated Positioning Method Based on UWB/IMU/ODOM/LIDAR

The map construction method based on IMU/ODOM/LIDAR is mainly to integrate the estimated pose information of IMU/ODOM via the front-end input, and the back-end scans the surrounding environment using LIDAR to obtain the positioning information in the environment. Lastly, the environment map is generated based on SLAM algorithm [20,21,22]. The above method has been found to be suitable for the indoor environment with flat ground, and it is highly dependent on the accuracy of sensors. The ground conditions of greenhouses are relatively complex, and the working area is generally wide. The accumulative error generated by ODOM and IMU will affect the accuracy of mapping and positioning. The integrated positioning method shown in Figure 2 uses UWB, IMU, ODOM, and LIDAR to address these issues.

The motion model and measurement model of greenhouse robot in the actual motion process are nonlinear. The Extended Kalman filter (EKF) algorithm can successfully cope with nonlinear systems, making it ideal for integrating numerous sensor inputs and estimating the relative pose of the robot [23]. Based on the particle filter algorithm, the Adaptive Monte Carlo (AMCL) is capable of fusing LIDAR information and map information for matching to monitor the global pose of the robot, and it exhibits a high degree of robustness. This makes it ideal for global positioning of the robot and navigation [24]. The combination of EKF and AMCL can allow them to complement each other for more accurate global positioning. Essentially, this paper's integrated positioning method is broken down into two stages. UWB/IMU/ODOM are initially integrated using the EKF algorithm to offer self-estimated robot platform pose information. EKF covers two steps, including prediction and measurement. The system uses the motion state information $\left(V_{X,O},V_{Y,O},\omega_{O}\right)$ provided by ODOM as the input of the prediction stage, while applying the yaw angle $\theta_{\mathrm{IMU}}$ provided by IMU and the absolute coordinate information $\left(X_{\mathrm{UWB}},Y_{\mathrm{UWB}}\right)$ provided by UWB as the input of the measurement stage to obtain the estimated pose information $\left(X_{\mathrm{EKF}},Y_{\mathrm{EKF}},\theta_{\mathrm{EKF}}\right)$ of the robot platform. There are additionally two phases in the AMCL algorithm: prediction and measurement. At the second stage, the estimated pose information obtained at the first stage is the input of the prediction, and the pose information $\left(X_{L},Y_{L},\theta_{L}\right)$ obtained by matching the LIDAR with the pre-established 2D grid map of the greenhouse serves as the input of the measurement stage [25], so as to achieve the global positioning of the robot in the greenhouse.

### 2.3. State Space Model

Assuming that the working area of the robot platform in the greenhouse is an ideal horizontal 2D environment, the state vector of the system is the pose of the robot, and the state vector of the robot platform at time *t* is $x_{t}={\left[X_{t},Y_{t},\theta_{t}\right]}^{T}$, where $X_{t}$ and $Y_{t}$ denote the position of the geometric center of the robot platform, and $\theta_{t}$ represents the yaw angle of the robot platform in the navigation coordinate system. The motion model and measurement model of the motion system are developed as follows using EKF and AMCL algorithms based on filtering [26]:(1)$$\{\begin{array}{c} x_{t}=g(u_{t},x_{t-1})+\varepsilon_{t}\\ z_{t}=h(x_{t})+v_{t} \end{array}$$
where $x_{t}$ denotes the state quantity of the system at time *t*; $u_{t}$ represents the control quantity of the system; $\varepsilon_{t}$ expresses the motion noise of the system; $z_{t}$ is the measured value of the system at time *t*; $v_{t}$ denotes the measured noise of the system.

#### 2.3.1. EKF Algorithm Integrates UWB/IMU/ODOM Positioning Data

According to the dead reckoning approach and the differential motion model of the mobile robot, the robot can be regarded as moving at a uniform speed in a short time [27]. The pose information $\left(V_{X,O},V_{Y,O},\omega_{O}\right)$ of the ODOM serves as the input of the control quantity in the prediction stage, and the pose of the robot platform at time *t* is expressed as:(2)$$\bar{x_{t}}=x_{t-1}+(\begin{array}{ccc} \cos\theta_{t-1} & -\sin\theta_{t-1} & 0\\ \sin\theta_{t-1} & \cos\theta_{t-1} & 0\\ 0 & 0 & 1 \end{array})(\begin{array}{c} V_{X,O}\\ V_{Y,O}\\ \omega_{O} \end{array})dt$$
where $V_{X,O}$, $V_{Y,O}$, and $\omega_{O}$ denote the motion speed of the robot platform in the x-axis direction, y-axis direction, and yaw angle in the navigation coordinate system, respectively.

The covariance matrix of the system state vector at time *t* in the prediction stage is written as:(3)$$\bar{P_{t}}=f_{x}P_{t-1}f_{x}^{T}+f_{w}\varepsilon_{t}f_{w}^{T}$$
where $P_{t}$ denotes the covariance matrix of the state quantity $x_{t}$; $\varepsilon_{t}$ represents the covariance matrix of the motion noise at time *t*; $f_{x}$ and $f_{w}$ denote the Jacobian matrix of the motion model and the motion noise, respectively.

For greenhouse environment, the motion noise of ODOM measured in reference [26] is written as:(4)$$\varepsilon_{t}=(\begin{array}{ccc} 0.01 & 0 & 0\\ 0 & 0.01 & 0\\ 0 & 0 & 0.5 \end{array})$$

In accordance with EKF, it takes the yaw angle $\theta_{t,\mathrm{IMU}}$ of IMU and the 2D positioning information $\left(X_{t,\mathrm{UWB}},Y_{t,\mathrm{UWB}}\right)$ provided by UWB as the input of state measurement value at time *t*, and the measurement model is expressed as:(5)$$z_{t}=(\begin{array}{c} X_{t,\mathrm{UWB}}\\ Y_{t,\mathrm{UWB}}\\ \theta_{t,\mathrm{IMU}} \end{array})+v_{t}$$
where $z_{t}$ denotes the system measurement value at time *t*; $v_{t}$ represents the measurement noise at time *t*.

The calculated kalman gain coefficient is calculated as:(6)$$K_{t}=\bar{P_{t}}H^{T}{\left(H\bar{P_{t}}H^{T}+v_{t}\right)}^{-1}$$
where *H* denotes the Jacobian matrix of the measurement model; $v_{t}$ represents the covariance matrix of the measurement noise. Combined with the sensor accuracy given by IMU and UWB manufacturers, the Rosbag toolkit in ROS system is adopted to record the positioning data of robot platform. After the derived EKF formula is analyzed by software simulation [28], the optimal measurement noise is set as:(7)$$v_{t}=(\begin{array}{ccc} 0.05 & 0 & 0\\ 0 & 0.05 & 0\\ 0 & 0 & 0.1 \end{array})$$

Corrected state quantity:(8)$$x_{t}=\bar{x_{t}}+K_{t}(z_{t}-\bar{x_{t}})$$

The covariance matrix of the corrected state quantity is calculated as:(9)$$P_{t}=(I_{3}-K_{t}H)\bar{P_{t}}$$
where $I_{3}$ denotes a three-order identity matrix.

#### 2.3.2. AMCL Algorithm Fuses LIDAR Positioning Data

AMCL algorithm falls into four stages, including particle initialization, prediction, measurement, and resampling [29]. The implementation steps are elucidated below:(1).Particle initialization

Taking the estimated pose information $x_{0}={\left[X_{\mathrm{EKF}},Y_{\mathrm{EKF}},\theta_{\mathrm{EKF}}\right]}^{T}$ obtained at the initial time EKF as the input of the initial estimated pose, the number of randomly generated particles *n* = 300, the generated initial particle set is recorded as ${\left\{(x_{0}^{i},w_{0}^{i})\right\}}_{i=1}^{300}$, and the weight of each particle is equated with $w_{0}^{i}=1/300$.

(2).Prediction stage

With the estimated pose information $x_{0}={\left[X_{\mathrm{EKF}},Y_{\mathrm{EKF}},\theta_{\mathrm{EKF}}\right]}^{T}$ obtained by EKF algorithm as the input of the prediction stage, the state of particles at time *t* is updated as:(10)$${\bar{x}}_{t}=x_{t-1,\mathrm{EKF}}+\varepsilon_{t}$$

(3).Measurement stage

Measurement noise $v_{t}$ of LIDAR is known. The pose information $z_{L}={\left[X_{L},Y_{L},\theta_{L}\right]}^{T}$ obtained by matching the LIDAR with the pre-established 2D grid map of the greenhouse is employed as the input of the measurement stage, and the particle weight $w_{t}^{i}$ at time *t* is obtained:(11)$$w_{t}^{i}=f_{R}(z_{t,L}-\bar{x_{t}^{i}})w_{t-1}^{i}$$
where $f_{R}(\text{·})$ denotes the probability distribution function of measurement noise $v_{t}$.

The particle set after measurement stage update is ${\left\{(x_{t}^{i},w_{t}^{i})\right\}}_{i=1}^{300}$. The weights of all particles are updated and then normalized to obtain $\bar{w_{t}^{i}}$:(12)$$\bar{w_{t}^{i}}=w_{t}^{i}/\sum_{i=1}^{300}w_{t}^{i}$$

Through the weight and pose information of the latest particle set, the optimal pose estimation of the robot platform at time *t* can be yielded as:(13)$$x_{t}=\sum_{i=1}^{300}\bar{w_{t}^{i}}x_{t}^{i}$$
where $\bar{w_{t}^{i}}$ denotes the weight of particle *i* after updating at time *t*; $x_{t}^{i}$ represents the optimal pose estimation of particle *i* after updating at time *t*.

(4).Resampling

The particles are filtered in accordance with the weight of all particles. The particles with higher weight are closer to the real pose. The threshold of effective particle number is set as $N_{ff}$ = 20, and the number of high weight particles is calculated as $N_{eff}$:(14)$$N_{eff}=1/\sum_{i=1}^{300}{\left(\bar{w_{t}^{i}}\right)}^{2}$$

When $N_{eff}<N_{ff}$, the particles are resampled by KLD algorithm [30], and the updated particles are introduced into (2) prediction stage for cyclic calculation. Accordingly, the longer the robot platform moves in the map, the more the sensor positioning information is obtained, the more accurate the positioning is.

### 2.4. Layout of Experiment Site

The experiment was performed in a vegetable greenhouse at Yunyuan Scientific Research Base of Hunan Agricultural University, Changsha, Hunan Province, China. As depicted in Figure 3, the layout in the greenhouse consisted of four field ridges, three aisles between ridges and a longitudinal aisle. The width of the field ridges is 1 m, and the width of the aisle between ridges is 0.9 m. The above aisles were paved with cement. Vegetable crops with an average height of 0.4 m were planted on the field ridges. A square area of 14 m by 8 m was selected for the test site, and a baffle was adopted to separate the test area in the aisle between ridges. Four UWB base stations were set at the four vertices of the test square area, respectively, and then fixed at a height of 0.7 m above the ground with a support. A 2D positioning and navigation coordinate system was built (Figure 3) with base station A as the origin of the coordinate system. The reference trajectory and four inflection points, including a (1.80, 1.80) m, b (12.60, 1.80) m, c (12.60, 5.65) m, and d (1.80, 5.65) m (Figure 3), were preset in the test site. During the test, the remote monitoring platform was employed to remotely monitor and record the real-time data of the robot platform via Wi-Fi.

This paper employs the RMSE as a measure of positioning accuracy [31]. The formula corresponding to this measurement is as follows:(15)$$\mathrm{RMSE}=\sqrt{\frac{1}{N}\sum_{n=1}^{N}{\left(m^{n}-m_{0}^{n}\right)}^{2}}$$
where *N* denotes the total number of sample data; *n* represents the sample data; m represents the actual value of the sample data; $m_{0}$ represents the target value of the the sample data.

Cartographer algorithm [12], an algorithm based on graph optimization, primarily creates a local map by fusing multi-sensor data, and it achieves loop detection using the local map. This algorithm is capable of reducing a certain amount of accumulative error in the process of mapping, and it applies to the map construction of greenhouses.

## 3. Experiments and Results

### 3.1. Precision Comparison Experiment of Greenhouse Mapping and Positioning

In this paper, the navigation boundary was obtained by scanning the crop-row and the sidewall of greenhouses with LIDAR [32,33]. The robot platform was manually controlled to move in the cement aisle area of the greenhouse in accordance with the same moving track and completed the map construction. A combination of the EKF algorithm and the AMCL algorithm is utilized for combining the data in the IMU/ODOM/LIDAR integrated positioning method. Moreover, the map is constructed by integrating IMU/ODOM/LIDAR and UWB/IMU/ODOM/LIDAR with SLAM algorithm, respectively, as presented in Figure 4.

To compare the mapping accuracy of the two combinations of sensors in the greenhouse environment, as presented in Figure 4, we marked the five feature areas in the environment, and the feature areas include the width of the greenhouse and the center width of the aisle between adjacent field ridges. The feature areas of the 2D map were mapped using the RVIZ software of ROS system to determine the map measured value [34], which was compared with the actual measured value. The results are listed in Table 1. As indicated by the results, a maximum error of 0.11 m was detected in the feature areas of the combined IMU/ODOM/LIDAR mapping, suggesting that the map may have drifted and that the error accrues with time. The maximum error of the feature areas of the combined mapping based on UWB/IMU/ODOM/LIDAR was 0.03 m, significantly lower than that of the IMU/ODOM/LIDAR mapping method.

The constructed greenhouse grid map was loaded using the RVIZ software of ROS system, and the positioning data of the robot in the map was obtained by the AMCL algorithm. After the robot's pose was initialized, the robot platform was manually controlled to track the reference trajectory at the speed of 0.5 m/s and recorded the trajectory of the robot in various maps and the trajectory of the single UWB positioning in real-time, as depicted in Figure 5a. Compare the lateral error between the recorded trajectory and the reference trajectory, as depicted in Figure 5b.

The lateral error of various positioning techniques is statistically and analytically calculated, and the findings are summarised in Table 2. The lateral average error of UWB positioning is 0.047 m, the maximum lateral error is 0.157 m, and the RMSE is 0.051 m. The maximum lateral error of the IMU/ODOM/LIDAR integrated positioning method is more than 0.2 m, and the RMSE is 0.103 m. The lateral average error of UWB/IMU/ODOM/LIDAR integrated positioning method in this paper is 0.027 m, and the maximum error is less than 0.1 m. The lateral RMSE is lowered by 33.3% and 67% when compared to UWB positioning and IMU/ODOM/LIDAR integrated positioning methods.

Based on the multi-sensor integrated method in reference [35], the EKF algorithm is adopted to integrate UWB/IMU/ODOM/LIDAR to compare the integrated positioning accuracy between the integrated algorithm in this paper and the conventional EKF algorithm. In the experiment, the robot platform adopts the single EKF algorithm and the EKF/AMCL combined algorithm to track the reference trajectory at 0.5 m/s, and the recorded lateral error results are presented in Figure 6. The lateral errors of the two integrated algorithms are analyzed, and the results are listed in Table 3. As revealed by the results, the maximum lateral error of single EKF algorithm is higher than 0.1 m, and the lateral RMSE of EKF/AMCL combined algorithm is reduced by 22.7% compared with single EKF algorithm.

The lateral error of UWB/IMU/ODOM/LIDAR integrated positioning method is recorded at three moving speeds to verify the positioning accuracy of the positioning method in this paper at different speeds, and the results are presented in Figure 7. The lateral error of the robot platform in different moving speeds is analyzed, and the results are listed in Table 4. As revealed by the results, the lateral error increases slowly with the increase of the speed. At 0.7 m/s, the average lateral error is obtained as 0.036 m, the maximum error is 0.095 m, and the lateral RMSE is 0.04 m.

### 3.2. Target Points Positioning Experiment

As part of this experiment, we compared the performance of the positioning system, the IMU/ODOM/LIDAR integrated positioning method, and the single UWB positioning. Twenty-four target points were set on the reference trajectory. The robot platform was manually controlled to track the reference trajectory, and guarantee that the center point of the robot platform coincides with the target point every time it reaches a target point. Every time the robot platform reaches a target point, halt and record the positioning data of different positioning methods at the current time for ten times and take the average value, and the positioning results are as depicted in Figure 8. As indicated by the results, the positioning accuracy of UWB in the greenhouse environment was between 0.04~0.23 m, whereas the positioning data fluctuated and was not consistent. The IMU/ODOM/LIDAR integrated positioning method produces accumulative error throughout robot platform movement, but its data continuity is strong, and the fluctuation is modest. An integrated positioning strategy using UWB/IMU/ODOM/LIDAR in conjunction with the other two positioning methods presented in this study provided the most accurate positioning results.

The target points positioning experiment data was analyzed, and the results are listed in Table 5. There is 45.5% and 41.5% improvement in overall positioning accuracy when using the UWB/IMU/ODOM/LIDAR integrated positioning method in comparison to using the single UWB positioning and the IMU/ODOM/LIDAR integrated positioning method, respectively. The RMSEs of x-axis direction, y-axis direction and overall positioning were found as 0.092, 0.069, and 0.079 m, respectively, and the maximum positioning error was 0.102 m. These results show an improvement in robot platform precision in the greenhouse.

### 3.3. Analysis of System Positioning Time

In the experiment, the respective positioning time of 1000 frames of data of the positioning system is recorded, as shown in Figure 9. The results suggest that the average positioning time of the system is 72.1 ms, and the longest positioning time is less than 80 ms.

## 4. Conclusions

This paper combines UWB positioning technology with slam mapping technology based on 2D LIDAR. Additionally, the use of EKF and AMCL algorithm for multi-sensor fusion develops the integrated indoor positioning system of greenhouse robot based on UWB/IMU/ODOM/LIDAR. The results have demonstrated that this method provides higher precision positioning for greenhouse robots.

The main research conclusions were drawn:UWB/IMU/ODOM/LIDAR-based integrated positioning method is proposed in this study. First, the estimated pose information is obtained by EKF integrating the positioning data of UWB/IMU/ODOM. On this basis, the 2D map of the greenhouse was created by scanning crop-rows with LIDAR. Second, AMCL integrated the LIDAR and map information to achieve global positioning of the greenhouse robot, which was accomplished.The precision comparison experiment results of greenhouse mapping and positioning demonstrate that the UWB/IMU/ODOM/LIDAR integrated positioning method in this paper improves the mapping and positioning accuracy compared with the IMU/ODOM/LIDAR integrated positioning method extensively used by conventional indoor mobile robots. At different moving speeds, the lateral error of the positioning method in this paper increases slowly over speed. At 0.7 m/s, the maximum error is 0.095m and the lateral RMSE is 0.04 m. The experimental results of target points positioning indicate that the positioning accuracy of UWB/IMU/ODOM/LIDAR integrated positioning method in this paper increased by 45.5% and 41.5%, respectively, compared with single UWB positioning and IMU/ODOM/LIDAR integrated positioning method. The RMSEs of x-axis direction, y-axis direction, and overall positioning are obtained as 0.092, 0.069, and 0.079 m, respectively, the maximum positioning error is 0.102 m, and the average positioning time of the system is 72.1 ms, thus meeting the positioning accuracy and positioning time requirements of robot navigation in greenhouse operation. Comparing the above test results to the results of existing studies [8,18,36,37], the positioning system proposed in this paper provided a higher level of positioning accuracy.

Since there may be irregular crops and irregular planting gaps in non-standard greenhouses, LIDAR has difficulty in obtaining a clear navigation boundary, which affects the accuracy of positioning and navigation [38]. The point cloud data scanned by LIDAR should be analyzed in depth. In addition, the multi-sensor fusion positioning method proposed in this paper was confirmed to be suitable for the relatively open greenhouse environment where dwarf crops are planted, and it could be more dependent on UWB to provide reliable location information. In further research, the greenhouse environment with tall crops will be investigated.

## Figures and Tables

**Figure 1 sensors-22-04819-f001:**
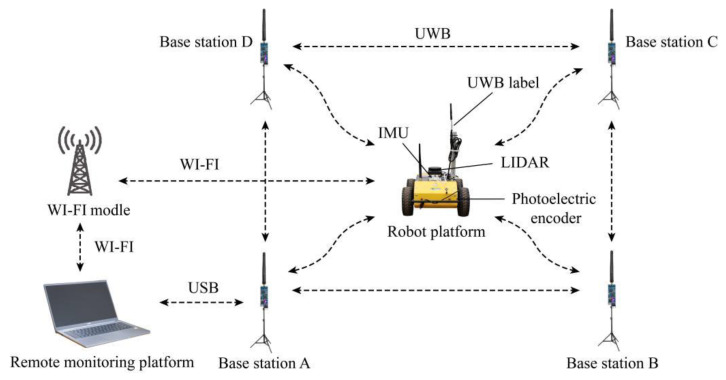
Schematic diagram of the positioning system.

**Figure 2 sensors-22-04819-f002:**
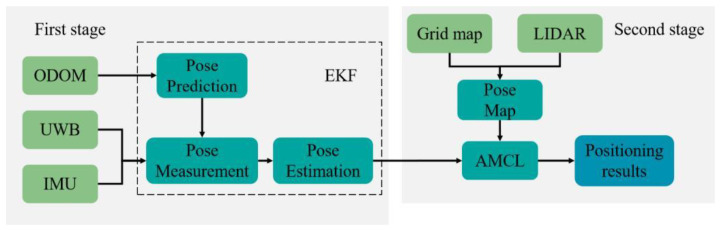
Integrated positioning frame diagram based on UWB/IMU/ODOM/LIDAR.

**Figure 3 sensors-22-04819-f003:**
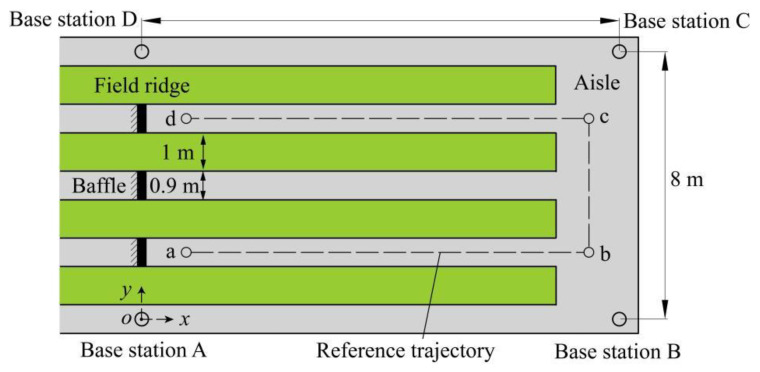
Schematic diagram of greenhouse experiment site.

**Figure 4 sensors-22-04819-f004:**
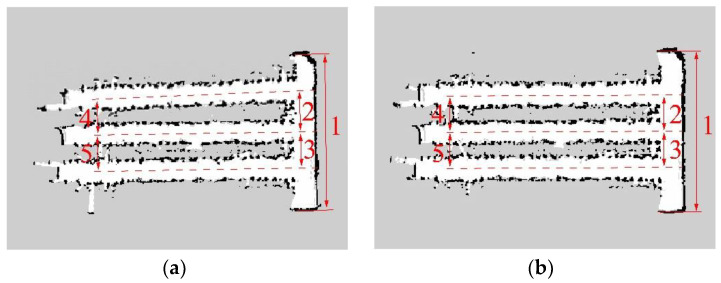
Greenhouse mapping with different combinations of sensors. (**a**) IMU/ODOM/LIDAR. (**b**) UWB/IMU/ODOM/LIDAR.

**Figure 5 sensors-22-04819-f005:**
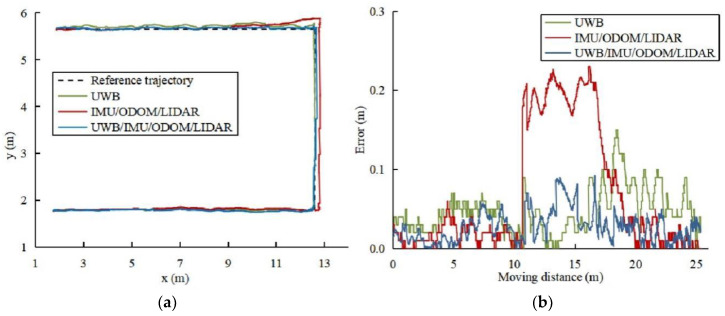
Compare the trajectories of different positioning methods. (**a**) Trajectories. (**b**) Lateral error.

**Figure 6 sensors-22-04819-f006:**
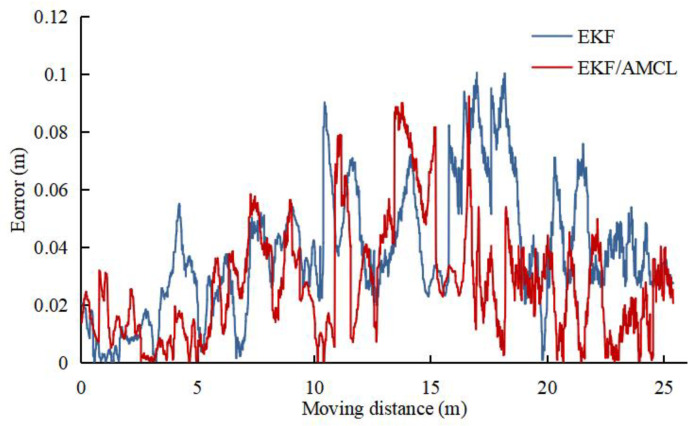
Comparison of lateral error between two integrated algorithms.

**Figure 7 sensors-22-04819-f007:**
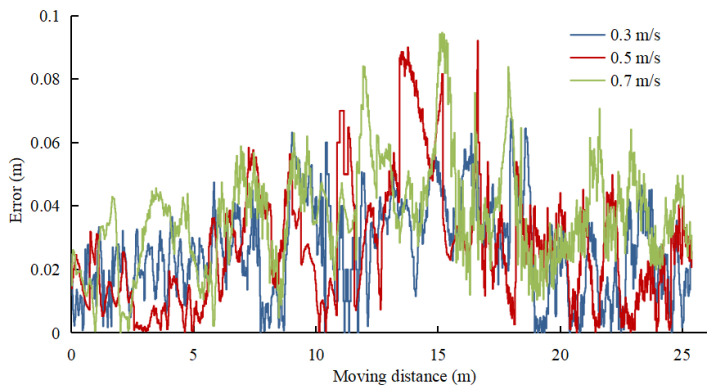
Lateral error at different moving speeds.

**Figure 8 sensors-22-04819-f008:**
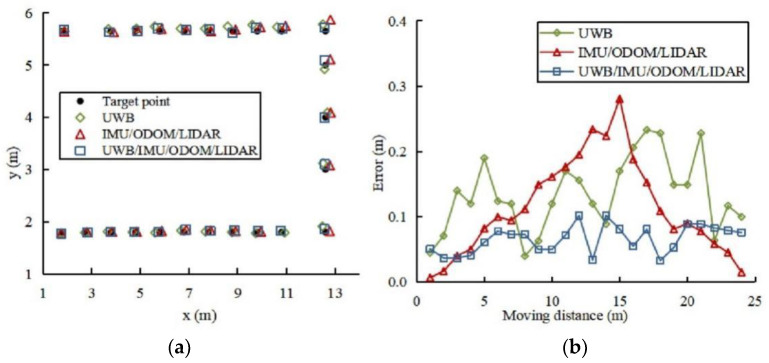
Different positioning methods compared for accuracy of target point positioning. (**a**) Positioning results. (**b**) Positioning error.

**Figure 9 sensors-22-04819-f009:**
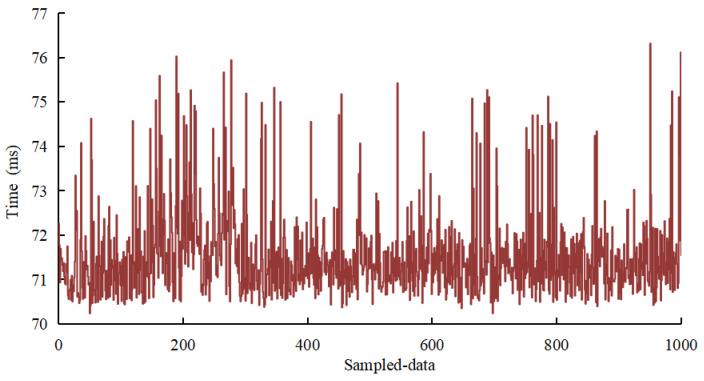
Positioning time.

**Table 1 sensors-22-04819-t001:** Comparison of mapping error of different combinations of sensors.

Feature Area	Actual Measured Value (m)	Map Measured Value (m)	
		IMU/ODOM/LIDAR	UWB/IMU/ODOM/LIDAR
1	8.40	8.46	8.43
2	1.90	1.96	1.89
3	1.90	1.89	1.88
4	1.90	1.79	1.90
5	1.90	1.85	1.88

**Table 2 sensors-22-04819-t002:** Statistics and analysis of lateral error of different positioning methods.

Positioning Method	Average Error (m)	Maximum Error (m)	RMSE (m)
UWB	0.047	0.157	0.051
IMU/ODOM/LIDAR	0.067	0.234	0.103
UWB/IMU/ODOM/LIDAR	0.027	0.091	0.034

**Table 3 sensors-22-04819-t003:** Analysis of lateral error of two integrated algorithms.

Integrated Algorithm	Average Error (m)	Maximum Error (m)	RMSE (m)
EKF	0.039	0.101	0.044
EKF/AMCL	0.027	0.091	0.034

**Table 4 sensors-22-04819-t004:** Statistics and analysis of lateral error at different moving speeds.

Moving Speed	Average Error (m)	Maximum Error (m)	RMSE (m)
0.3 m/s	0.021	0.067	0.03
0.5 m/s	0.027	0.091	0.034
0.7 m/s	0.036	0.095	0.04

**Table 5 sensors-22-04819-t005:** Error analysis of target points positioning with different positioning methods.

Positioning Method	RMSE (m)			Overall Maximum Error (m)
	x-Axis Direction	y-Axis Direction	Overall	
UWB	0.140	0.083	0.145	0.233
IMU/ODOM/LIDAR	0.127	0.072	0.135	0.281
UWB/IMU/ODOM/LIDAR	0.092	0.069	0.079	0.102

## Data Availability

All data are presented in this article in the form of figures and tables.

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
