# Peer review of "Integrated Indoor Positioning System of Greenhouse Robot Based on UWB/IMU/ODOM/LIDAR"

_sensors, 2022, doi:10.3390/s22134819_

Round 1

Reviewer 1 Report

This paper proposes an integrated positioning method for the greenhouse robot system  based on UWB/IMU/ODOM/LIDAR. In specific, both the classical EKF and the particle filtering methods are employed to fuse the UWB/IMU/ODOM/LIDAR measuring data, thus the proposed positioning method can correct the accumulative positioning error and achieve better positioning performance than the existing methods. Moreover, experiments are conducted. Yet, there are some problems in the paper:

1.  The proposed positioning method simply applies both the classical EKF and the particle filtering to the positioning or tracking method. However, there are no comments on the advantages about the proposed method and no comparison between EKF or PF based method and the proposed one.  Moreover, the conclusion is trivial that the more sensing data the positioning method uses, the better positioning performance it achieves.(such as the proposed method with UWB/IMU/ODOM/LIDAR  data  vs  the conventional method with IMU/ODOM/LIDAR  data)

2. There is lot of literature in which multi-sensor data is fused to improve the positioning performance. It would be convincing if the authors can compare their method with existing work.

Reviewer 2 Report

The paper illustrates with a real testbed the performance of a sensor and data fusion approach to the positioning of mobile robots in greenhouses. The approach followed by the authors is to combine data coming from UWB, IMU, ODOM, and LIDAR mechanisms into an integrated two-stage positioning solution based on an Extended Kalman Filter and a Particle Filter, respectively.

Basically, the scenario is open-space and in the radio line-of-sight conditions, which greatly simplifies achieving optimal performance in terms of localization accuracy. Although this is compatible with the reference application scenario, a slightly more challenging radio environment should be considered, for instance by adding obstacles or placing the base stations closer to the ground.

The results are only focused on the localization error, but no performance analysis is provided: how fast is the system to produce an accurate result? This also translates into a probable limitation on the maximum speed at which the robot can move and, especially turn, on the playground. Some results on this matter and tests with different speeds of the robot should be included too and presented individually as done for the different combinations already shown in Figs 5 and 6.

Besides these major aspects, there are some further minor issues that deserve attention, as listed below.

1. On Lines 58-59, the authors claim that the approach proposed in [12] is very accurate but has a high cost, because of the dual-LIDAR usage. In these terms, the authors should spend some more words evaluating the cost related to their proposed solution, since one extra LIDAR is substituted by a bunch of other hardware components which reduces a little the impact of their comment.

2. On lines 73-75, the authors claim that UWB, as the only location algorithm, is easily blocked by obstacles and interference. This is untrue since, thanks to its nature of ultra-wide bandwidth, the obstacles-penetration capabilities are much higher than other radio technologies. Indeed, in the greenhouse, the radio environment is expected to be quite closer to full line-of-sight conditions, as also demonstrated by the results in Tables 2 and 3, where UWB alone closely approaches the final RMSE error level. Accordingly, the authors should find a better justification when commenting on the paper [15] on the referred lines.

3. On line 214, the expression is formally wrong, in the sense that the authors expand the x_t but not w_t^i

4. On lines 277-279, the authors write that "The maximum error [..] was 0.03 m, significantly higher than [..]". It should be "lower" not "higher".

Reviewer 3 Report

The article deals with the application of robots in the greenhouse environment for growing agricultural crops. This application is very interesting and very necessary. In this area of ​​application, there is still room for robotics applications in the future. In some countries, there is an acute shortage of workers in this area, so the application of robots is a solution to this problem. For autonomous robot applications, it is also necessary to address the global positioning of the robot and its navigation in this environment. The article addresses the methodology of robot localization and navigation. I find the article interesting and highly topical.

The greenhouse environment has specific conditions for the application of environmental mapping techniques and it is therefore necessary to address the application possibilities of individual available technologies for this purpose and to verify their applicability experimentally.

In the introduction, the authors compare the various methods of robot localization with reference to individual relevant references. A robot with a four-wheel drive and differential steering was used for the application. The robot is equipped with an odometer, UWB label, LIDAR and IMU and has encoder sensors in the wheels to determine the position of the robot. The data fusion system and their processing is further designed. In the next step, an experiment is proposed to verify the authors' suggestions. The authors proposed a methodology for solving these problems and subsequently verified it experimentally with excellent results.

Comments:

in Table 1, the second line is incorrect: "189".

Figures 5 and 6 have poor resolution.

Round 2

Reviewer 1 Report

1. Response to comment: (The proposed positioning method simply applies both the classical EKF and the particle filtering to the positioning or tracking method. However, there are no comments on the advantages about the proposed method and no comparison between EKF or PF based method and the proposed one. Moreover, the conclusion is trivial that the more sensing data the positioning method uses, the better positioning performance it achieves. (such as the proposed method with UWB/IMU/ODOM/LIDAR data vs the conventional method with IMU/ODOM/LIDAR data) )

The Author’s Response: Thank you for your comments. We have added the advantages of the proposed method in lines 144-147 of the new manuscript. The method proposed in this paper is mainly based on EKF and AMCL for fusion positioning. AMCL is the algorithm framework in slam system, which cannot complete positioning alone and it needs to input the robot's state prediction information and LIDAR information. Therefore, we use EKF to provide state prediction information (estimated pose information) for robot. We introduce UWB into the LIDAR-SLAM positioning system and propose a UWB/IMU/ODOM/LIDAR integrated method. And experimentally found that UWB can eliminate the positioning accumulative error of the traditional method and improve the accuracy of slam mapping. In the introduction, we also commented on UWB technology that it is not possible to increase the number of sensors arbitrarily to improve the positioning accuracy. From figures 2 and 4, it can be seen that the positioning accuracy of UWB alone is not much lower than that of IMU/ODOM/LIDAR method, and these two positioning methods have their own characteristics. The proposed method in this paper combines the two positioning technologies to obtain higher positioning accuracy, while improving the accuracy of slam mapping in greenhouse, which can be a reference for existing relevant researchers.

 The reviewer’s Response: Both the EKF based positioning method and the Particle Filtering(PF) based positioning method can be applied to the Indoor Positioning System of Greenhouse Robot. In other words, the PF based method can realize the positioning or tracking for the Greenhouse Robot system. Please compare the proposed method with the classical EKF or PF based positioning or tracking method for this Greenhouse Robot System to show the advantages of the proposed one, since the proposed one simply applies both the classical EKF and the particle filtering to the positioning or tracking method. 

2. Response to comment: (There is lot of literature in which multi-sensor data is fused to improve the positioning performance. It would be convincing if the authors can compare their method with existing work.)

The Author’s Response: Thank you for your comments. At present, the multi-sensor fusion positioning technology based on LIDAR is mainly applied indoors, with less research in the greenhouse environment. The method widely used by indoor mobile robots is IMU/ODOM/LIDAR fusion for positioning, which has the problem of positioning accumulative error in the greenhouse. We have added relevant comments and references in lines 62-65 of introduction of the new manuscript. In the experiment, we use the integrated positioning frame in this paper for integrated positioning for both the proposed UWB/IMU/ODOM/LIDAR method and the traditional IMU/ODOM/LIDAR method. In addition, we have compared the existing technologies in the experiments, such as UWB, IMU/ODOM/LIDAR. We have also compared the accuracy of existing positioning technologies in lines 376-378 of the conclusion of the manuscript.

The reviewer’s Response: The existing multi-sensor fusion positioning technology based on LIDAR can be applied to the greenhouse environment. Please compare the existing method with the proposed method in the greenhouse environment. 

Reviewer 2 Report

In this version, most of my previous comments have been properly addressed and I think this contribution has improved enough.

However, I have a substantial remark about the way the authors are justifying the need for their proposed approach. One of the issues raised was the fact that UWB alone is demonstrated to closely approaches the final RMSE error level. Nevertheless, the authors are not giving a proper justification for the need to further improve the accuracy achieved by UWB alone to meet the requirements of greenhouse robot operations. 

For instance, I believe the way the authors dealt with my comment on reference [15] (now [17]) is not in line with what I asked them to do and now the comment to that reference got worse. In fact, in this version of the paper, they are claiming that an accuracy level ranging from 7 cm to 8 cm is not enough to meet the positioning requirements of the reference application scenario.

In my opinion, this accuracy level might be more than enough for the referenced application scenario, but if the authors think differently, they should provide clear pieces of evidence on why and in which circumstances the accuracy achieved by UWB alone is not enough for greenhouse robot operations.

If they fail to do so, it's natural to object that it is this one the proposal that leads to additional economic (due to extra hardware) and processing (due to extra computations and eventually delays) costs, which are not properly justified in the addressed application scenario.

This issue is tightly coupled with the other one that I mentioned about making the scenario more challenging by including obstacles and further radio impairments. Nevertheless, the authors opted to leave this to their future work, so, at present, the feeling is that there is not enough material to clearly justify the need for this approach.

Round 3

Reviewer 1 Report

The author has responsed to my review comments. I have no more comments on this revised manuscript.

Reviewer 2 Report

This new version of the manuscript finally addressed all my previous comments in a satisfactory way, where some more meaningful justifications were given about the concerns raised.

For the specific application field, I'm still convinced that in terms of practical use, the accuracy level reached by the UWB-only approach might be enough. This is because a robot itself is not a point spatial object but has a dimension, whose base might be 50 cm x 50 cm (or even bigger) and the error levels demonstrated by the UWB-only approach almost never lead to locating the robot far away from where it actually is.

Nevertheless, I have no more major objections to the proposed approach being accepted as an improvement to the overall localization accuracy.

This manuscript is a resubmission of an earlier submission. The following is a list of the peer review reports and author responses from that submission.

Round 1

Reviewer 1 Report

For an expression (2), if ‘dt’ is a differential of the variable ‘t’, the mentioned expression (2) should have an integral from a motion speed and yaw angle speed.

For Fig.4, what is the difference between images (a) and (b)? It seems that system UWB/IMU/ODOM/LIDAR or IMU/ODOM/LIDAR does not depend on UWB and IMU signals. Perhaps, the system depends from ODOM and LIDAR signals mainly.

For an expression (3), what is denoted as ‘fw ?

For the References, all sources, except No.3 and No.10, are mentioned authors from South-East Asia. It would be nice to add more sources from other sites of the world.

Reviewer 2 Report

This paper proposes an integrated positioning method for the greenhouse robot system  based on UWB/IMU/ODOM/LIDAR. In specific, both the classical EKF and the particle filtering methods are employed to fuse the UWB/IMU/ODOM/LIDAR measuring data, thus the proposed positioning method can correct the accumulative positioning error and achieve better positioning performance than the existing methods. Moreover, experiments are conducted. Yet, there are some problems in the paper:

  1. The proposed positioning method simply applies both the classical EKF and the particle filtering to the positioning or tracking method. However, there are no comments on the advantages about the proposed method and no comparison between EKF or PF based method and the proposed one. Moreover, the conclusion is trivial that the more sensing data the positioning method uses, the better positioning performance it achieves.
  2. There is literature in which multi-sensor data is fused to improve the positioning performance. It would be convincing if the authors can compare their method with existing work.
  3. The function “fR()” in the equation (11) is not defined.
  4. Except for the proposed positioning method, i.e. UWB/IMU/ODOM/LIDAR based positioning method, the positioning methods, such as the IMU/ODOM, IMU/ODOM/LIDAR, UWB, are not mentioned in this paper.
  5. There is accumulative positioning error in the IMU/ODOM based method and the IMU/ODOM/LIDAR based method, while the positioning errors of the IMU/ODOM based method and the IMU/ODOM/LIDAR based method in fig.6 are not increasing in the tracking phase. More explanation is needed for this.

Reviewer 3 Report

The paper is very well written, easy to follow, and interesting. I hope that subsequent papers will be written to show the use of such combination of sensors on more irregular objects, but this is a good first publication. The technical aspects of the paper and the writing were excellent. The comparison studies helped the reader see how the LiDAR introduced to the system made a difference.

One small writing suggestion. Page 9: Sentence starting with 24 should use words "Twenty-four" since the number starts a sentence.